# Metabolic Alterations in Cellular Senescence: The Role of Citrate in Ageing and Age-Related Disease

**DOI:** 10.3390/ijms23073652

**Published:** 2022-03-26

**Authors:** Maria Elzbieta Mycielska, Emma Naomi James, Eric Kenneth Parkinson

**Affiliations:** 1Department of Structural Biology, Institute of Biophysics and Physical Biochemistry, University of Regensburg, Universitätsstrasse 31, D-93053 Regensburg, Germany; maria.mycielska@biologie.uni-regensburg.de; 2Centre for Oral Immunobiology and Regenerative Medicine, Institute of Dentistry, Barts and the London School of Medicine and Dentistry, Queen Mary University of London, Turner Street, London E1 2AD, UK; dremmajames@gmail.com

**Keywords:** senescence, telomere, metabolism, citrate, ageing, cancer, transport, energy

## Abstract

Recent mouse model experiments support an instrumental role for senescent cells in age-related diseases and senescent cells may be causal to certain age-related pathologies. A strongly supported hypothesis is that extranuclear chromatin is recognized by the cyclic GMP–AMP synthase-stimulator of interferon genes pathway, which in turn leads to the induction of several inflammatory cytokines as part of the senescence-associated secretory phenotype. This sterile inflammation increases with chronological age and age-associated disease. More recently, several intracellular and extracellular metabolic changes have been described in senescent cells but it is not clear whether any of them have functional significance. In this review, we highlight the potential effect of dietary and age-related metabolites in the modulation of the senescent phenotype in addition to discussing how experimental conditions may influence senescent cell metabolism, especially that of energy regulation. Finally, as extracellular citrate accumulates following certain types of senescence, we focus on the recently reported role of extracellular citrate in aging and age-related pathologies. We propose that citrate may be an active component of the senescence-associated secretory phenotype and via its intake through the diet may even contribute to the cause of age-related disease.

## 1. Introduction

### 1.1. Cellular Senescence

Classical cellular senescence was originally defined as an irreversible cell cycle arrest that was distinct from quiescence, terminal differentiation, and apoptosis. This form of senescence could occur following multiple rounds of cell division (replicative senescence [1]) or following a wide range of cellular stresses such as ionizing radiation, signaling imbalance, and oxidative damage (stress-induced premature senescence or SIPS [2]). However, more recently the definition of senescence has been broadened to include developmental senescence, oncogene-induced senescence, and cancer-therapy-induced senescence where an irreversible cell cycle arrest is either unstable [3] or does not occur at all [4]. The process of irreversible senescence takes some time to complete following its induction and for some time can be transiently rescued by the disabling of p53 [5], p38 mitogen-activated kinase [6], a combination of p16^INK4A^ and p21^WAF^ [7] or EGR2 [8]. In addition, confluent or serum-starved quiescent cells eventually develop DNA damage, elevated p21^WAF^ and p16^INK4A^, and transit into senescence [9,10], suggesting that senescent cells may result from an initially reversible quiescent state. In addition, a phenotype resembling cellular senescence occurs in post-mitotic tissues such as the liver and brain, following cellular stress and during chronological aging [11,12] but this is now thought to be a distinct phenotype and is termed ‘Long-lived post mitotic ageing’ see [11] for a recent review.

### 1.2. The Senescence—Associated Secretory Phenotype (SASP)

The SASP is a collection of proteins secreted by a variety of senescent cell types and includes a large number of cytokines, chemokines and immunomodulatory molecules, growth factors, shed surface molecules, and survival factors together with promoters of angiogenesis, fibrosis, and tissue remodeling. The SASPs are to some extent cell type- and mechanism-specific and recent data using single-cell RNA sequencing indicates that the SASP may be temporally and highly heterogeneous in nature even within populations of the same cell type [13]. The evolution of the classical senescent phenotype is a dynamic process and the upregulation of p21^CIP1/WAF1^ is an early consequence of telomere attrition and/or the induction of genotoxic stress, so some of the heterogeneity seen in senescent fibroblast populations could be due to the rate of progression of each cell towards senescence. Nevertheless, some of the cytokines of the SASP increase to detectable levels in the plasma of older humans and are proposed to be biomarkers of age-related conditions, such as frailty [14]. This sterile inflammation is known as inflammaging [15] and is thought to be due to the accumulation of senescent cells which display inappropriate levels of cytoplasmic chromatin which then detected by the cyclic GMP–AMP synthase (cGAS)–stimulator of interferon genes (STING) (cGAS-STING) pathway induce the inflammatory cytokines [16]. However, many centenarians have detectable levels of some SASP cytokines [15] and it has been hypothesized that they may adapt by increasing the levels of inhibitory cytokines such as IL-10 [17].

### 1.3. Telomeres and Senescence

Replicative senescence occurs at the end of multiple rounds of replication known as the Hayflick limit [1] and is strongly linked to the gradual erosion of the repeat DNA sequences and their associated proteins (the shelterin complex [18,19]) at the end of the chromosomes (the telomeres) in cells which lack the enzyme telomerase. One of the main tasks of telomeres is to prevent the natural ends of human chromosomes from engaging in end-to-end fusions or being misinterpreted as DNA double-strand breaks. Telomere attrition is thought to be due at least in part to the end replication problem pointed out many years previously [20]. Telomerase is present in germ cells, stem cells, and the highly proliferative cells of epithelia and parts of the hematopoietic system, but is absent or low in abundance in all other cells [21]. For telomere attrition to occur cells must be telomerase deficient and divide [22]. In telomerase-deficient cells in vitro telomeres shorten until at least five of them are short enough to trigger cellular senescence [23]. Short telomeres fail to assemble the protective shelterin complex, which in turn leads to the exposed DNA ends being recognized as DNA double-strand breaks [24,25] and cell cycle arrest ensuing through activation of checkpoint kinases, p53 and p21^WAF1/CIP1^. These exposed DNA ends can be visualized by their association with numerous DNA damage recognition proteins such as ᵞH2.AX and 53BP1 [24,25,26].

Telomere loss [27,28] and the inhibition of telomerase activity [29,30] in dividing cells can also occur when cells are subjected to high levels of oxidative damage but there is some evidence that this form of telomere loss can be reversed in the presence of sufficient levels of telomerase [3,31]. Furthermore, telomerase activity is regulated by numerous hormones and growth factors [32]. Indeed age-adjusted telomere attrition in leukocytes has been suggested as an indicator of human health [33,34] and is reversible upon positive changes in lifestyle [35,36] or increased telomerase activity [37]. Therefore, it is not clear whether leukocyte telomere shortening during human aging is due to a direct effect of reactive oxygen species on telomeres [27,38], the inhibition of telomerase activity [29,30], both or neither.

Additionally, telomeres can contribute to senescence and aging in other ways. More recently it has been shown that telomeres are also especially sensitive to the induction of DNA double-strand breaks by oxidative damage due to their high G:C content and poor DNA repair capacity [27,38] as a result of their inability to engage non-homologous end joining (NHEJ) DNA repair [27]. This form of DNA damage then results in telomere-associated foci or TAFs [38]. In non-dividing cells, which lack telomerase and the ability to engage homologous recombination (HR) repair the DNA damage, and the associated TAFs persist and gradually accumulate during chronological aging [27,38]. It is therefore important to distinguish between telomere maintenance in dividing cells and in post-mitotic tissue [11].

### 1.4. Telomere Dysfunction Regulates Mitochondrial Function and ROS and Vice Versa

The effects of dysfunctional telomeres on aging and mitochondrial function have mainly been studied in the telomerase-deficient mouse where either the telomerase RNA template gene *Terc* or the catalytic subunit gene *Tert* is homozygously deleted and the telomeres allowed to shorten over four generations. Numerous age-related phenotypes ensue including premature greying of the hair, poor wound healing, increased cancer incidence, gut defects, infertility, shortened lifespan, decreased adipose tissue, and hair loss [39]. Werner’s Syndrome (WS) is another progeroid syndrome associated with defects in telomere maintenance and caused by mutations in the *WRN* gene, which encodes the RecQ type DNA helicase for the unwinding of unusual DNA structures. WS has some common features with the *Terc−/−* mouse [40]. The human condition dyskeratosis congenita (DC) is associated with heterozygous mutations in *TERT* and *TERC* and hemizygous mutations in the telomerase enzyme component dyskerin (*DKC*) [41]. However, some of the premature aging phenotypes seen in WS are absent in DC such as type II diabetes, osteoporosis, cataracts and atherosclerosis [40]. Unsurprisingly, the DC phenotype resembles more the phenotype of the heterozygous *Terc* +/− mouse than the homozygous knockout mouse [42]. Telomere length is slightly shorter in all DC tissues tested [43] and *DKC*-deficient human fibroblasts have a very short replicative lifespan in vitro [44]. However, unlike the telomerase deficient mouse, DC patients commonly present with abnormal skin pigmentation, nail dystrophy, bone marrow failure, and leucoplakia [41]. Other less common somatic features in DC are pulmonary disease, liver disease, premature hair loss and/or greying, malignancy, cardiomyopathy, and osteoporosis but not commonly type 2 diabetes [41].

In mice nullizygous for telomerase short telomeres give rise to mitochondrial dysfunction and increased mitochondrial ROS via p53 and peroxisome proliferator gamma co-activator 1 alpha (PGC1 alpha [45]) a phenotype that is protected by restoring telomerase [45] or by generating mice with extremely long telomeres [46]. Telomere attrition/dysfunction is known to be associated with DNA damage foci detectable by ᵞH2AX foci [24,25] and this would lead to p53 activation, mitochondrial dysfunction, and increased ROS [45]. It has also been suggested [45], based on studies in yeast [47], that telomere dysfunction could lead to mitochondrial dysfunction and genomic instability via a decline in iron-sulphur cluster biogenesis. Damaged or senescent cells also accumulate g-H2AX which form telomere dysfunction-induced foci and correlate with telomere shortening. Conversely, the formation of g-H2AX has been suggested to be due to the oxidative damage caused by mitochondrial disfunction leading in turn to p53-dependent cell cycle arrest and it has been hypothesized that mitochondrial dysfunction via the production of ROS can lead to telomere attrition or dysfunction and to premature aging [48,49] and this hypothesis is supported by the fact that both mitochondrial superoxide dismutase 2 (SOD2) [50] and PGC1 alpha [51] nullizygous mice show deleterious epidermal phenotypes. Therefore, whether senescence is induced first by telomere shortening resulting in mitochondrial dysfunction, or the converse, these two cellular events appear to be associated.

However, the above epidermal phenotypes are not so dramatic in human skin aging and recent data implicates aged melanocytes as the major senescent cell type in the human epidermis where telomere-associated DNA damage is induced in the keratinocytes by a paracrine mechanism [52]. However, in aged human skin in situ, this was mainly seen in the granular layer where the keratinocytes are deficient in telomerase [53,54]. Therefore, it is not clear whether these phenotypes are seen in telomerase-proficient human keratinocytes, as most of the in vitro work has so far been conducted using systems where keratinocyte telomerase activity is likely to be deficient or absent [55].

### 1.5. Dietary Factors and the Regulation of Nicotinamide Adenine Dinucleotide (NAD+), Telomere and Mitochondrial Dysfunction and Its Relationship to Senescence

Very recently, dietary factors have emerged which may also impact the differences between DC and telomerase-deficient mice as well as offering a potential explanation for some in vitro findings on senescent cell metabolism. Of particular interest has been nicotinamide mononucleotide (NMN) and nicotinamide adenine dinucleotide (NAD+), molecules central to normal metabolic function and to the maintenance of a normal redox state within the cell. Decreased NAD+ levels have been implicated in senescence. NAD+ is an important cofactor to sustain glycolytic activity and is necessary for the conversion of glyceraldehyde-3-phosphate to 1,3-bisphosphoglycerate. While conversion of glyceraldehyde-3phosphate is accompanied by reduction of NAD+ to NADH further conversion of pyruvate to lactate is accompanied by NADH oxidation. Whether pyruvate is converted into lactate or enters the tricarboxylic acid (TCA) cycle is determined by the pyruvate dehydrogenase complex (PDHC—see also below) but pyruvate can also act as a scavenger of reactive oxygen species such as hydrogen peroxide and contribute to antioxidant defense [56,57].

NAD+ has been shown to decline in several tissues with age while disruption in its biosynthesis has been implicated in age-related diseases. NR and NMN supplementation in the diet has been shown to be effective in restoring NAD+ levels and slowing down age-associated phenotypes [58]. NAD+ in senescent fibroblasts can be also replenished by the uptake of NR and NMN (precursors of NAD+) from the culture medium as both accumulate in senescent fibroblasts [59,60]. The phenotypes of DC fibroblasts and telomerase nullizygous mice can be rescued by the inclusion of nicotinamide riboside (NR; an NAD+ precursor) in the culture medium or the diet and inhibiting the NADase CD38 by a short hairpin RNA [61]. These phenotypes included improved NAD homeostasis, thereby alleviating telomere damage, defective mitochondrial biosynthesis and clearance, cell growth retardation, and cellular senescence [61], which has been implicated in the reduction of plasma NAD+ levels in humans with chronological age [62]. In addition, similar manipulations inhibit the STING pathway in ataxia telangiectasia-mutated cells in vivo and in vitro [63]. These studies link dietary factors with the regulation of many senescent phenotypes. It is therefore likely that many fetal bovine serum batches and mouse diets are deficient in the vitamin B3 metabolites necessary to maintain NAD+ levels. Nicotinamide itself and nicotinamide mononucleotide (NMN) do not decline in human blood with age [64] and so these dietary variables may be very important when considering how to translate mouse model and in vitro data into human clinical trials [65]. NR supplementation has recently been reported to increase the NAD+ metabolome in aged human skeletal muscle, without evidence of any change in mitochondrial bioenergetics or a change in muscle and whole-body metabolism [65]. However, NR supplementation did reduce the level of some circulating SASP inflammatory cytokines [65] reviewed in [66]. Furthermore, these issues will be very important to consider how current thinking on the role of telomeres and mitochondria impact human aging and age-related diseases.

## 2. Senescent Cell Metabolism

The changes in senescent cell metabolism have been extensively reviewed recently [56] and so only a brief description of the key points will be provided here.

### 2.1. Glycolysis and Pyruvate

It has long been known that cancer cells shift their metabolism away from oxidative phosphorylation towards glycolysis even in the presence of atmospheric oxygen and this process is termed ‘aerobic glycolysis’ or the ‘Warburg Effect’ although this process can shift back towards oxidative phosphorylation in cancer stem cells reviewed in [67]. Nevertheless, it has become apparent in recent years that senescent cells also shift their metabolism in a manner that resembles aerobic glycolysis [68,69,70] accompanied by a less energetic state and an increase in the AMP to ATP ratio which can also result in the induction of senescence [68]. Pyruvate, the end metabolite of glycolysis, can be either metabolized to lactate or be transferred to mitochondria and metabolized to Acetyl-CoA (acetyl coenzyme A) necessary for citrate synthesis in the Krebs cycle. Pyruvate, which is at the intersection of glycolysis and mitochondrial respiration, is also a key metabolite in senescent cell metabolism [51]. Synthesis of pyruvate in the process of glycolysis requires the reduction of NAD+ to NADH. The NAD+ pool can be replenished through the action of lactate dehydrogenase which produces lactate and NAD+ from pyruvate, the former of which is secreted and can accumulate outside senescent cells in some instances [51]. Normally, the NAD+/NADH balance is also maintained in mitochondria through the action of ME2 (malic enzyme) and MAS (malate-aspartate shuttle); malic enzyme couples mitochondria with aerobic glycolysis in osteoblasts [57]; however, mitochondrial dysfunction-associated senescence has been shown to have decreased the NAD+/NADH ratio [51]. NAD+ level is also maintained in senescent fibroblasts most likely by the uptake of NR and NMN from the culture medium as both accumulate in senescent fibroblasts [59,60].

### 2.2. Energy Regulation in Oncogene-Induced Senescence

PDHC (pyruvate-dehydrogenase complex) is the gatekeeping enzyme linking glycolysis to the TCA cycle. This step is regulated by reversible phosphorylation by PDK (pyruvate dehydrogenase kinase) enzymes (PDK1–4) that inhibit its activity. In contrast, dephosphorylation by PDP1 and PDP2 (pyruvate dehydrogenase phosphatase) stimulates PDHC activity. Oncogenes such as *RAS* and *BRAF* can increase the flux of pyruvate into the TCA cycle and mediate the initiation of senescence by increasing the oxygen consumption rate and redox stress [71]. This is achieved by increasing levels of PDP2 and decreasing the levels of PDK1 and these phenotypes were reversed upon abrogation of the senescent phenotype by enforced normalization of PDK1 through over-expression or the knockdown of PDP2 using shRNA [71]. Furthermore, depletion of PDK1 eradicated melanoma subpopulations resistant to RAF kinase inhibitors suggesting PDK1 might be a tractable therapeutic target for melanoma [71]. Conversely, increasing glycolytic flux by overexpressing the glycolytic enzymes phosphoglycerate mutase or glucosephosphate isomerase slowed senescence induced by both *RAS* over-expression and proliferative exhaustion [72]. Furthermore, increased glycolytic flux is also a feature of immortal embryonic stem cells [73]. Therefore, increased glycolytic flux can delay the initiation of senescence in certain settings.

### 2.3. Energy Regulation in Replicative Senescence

In fully senescent fibroblasts cultured in the presence of FBS optimized for the culture of normal human cells, most glycolytic and pentose phosphate pathway (PPP) metabolites and enzymes and their transcripts increase and many TCA metabolites decline. In particular, the enzyme transcripts of the PDHC such as PDKs 1-4, and especially PDK4, are upregulated [70]. In addition, the levels of PDP2 are unchanged supporting the metabolic profiling data and indicating that flux into the TCA cycle via PDHC is reduced; this is in stark contrast to the increase in PDHC activity seen during the early stages of oncogene-induced senescence [71] and the reasons for this are not clear. However, the dynamic nature of the senescent phenotype and the culture condition variables such as adequate levels of NMN or NR and the presence or absence of pyruvate, to name but two, may play a part and this needs to be addressed in future work (see Section 1.5 and below).

### 2.4. Unbiased Metabolomic Screens of Senescent Cells

In the two unbiased metabolic screens of senescent oral NHOF-1 oral and IMR90 embryo lung fibroblasts published so far, there are many similarities. Specifically, glucose together with glycolytic metabolites [59,60] and pdk1-4 transcripts [28,70] increase in senescent cells as do the PPP metabolites ribulose/xylulose 5-phosphate, ribose, and ribose 5-phosphate [59,60] and these changes are not seen in confluent or quiescent cells [60]. In senescent oral fibroblasts, the AMP to ADP ratio did increase specifically [60] as reported previously [60,68,70] consistent with a shift towards glycolysis [60,68,70]. The shift towards glycolysis is independent of high levels of ROS as senescent NHOF-1 cells do not show a high oxidized/reduced glutathione ratio [60,70], whereas senescent IMR90 cells [59] do. Senescent NHOF-1 cells specifically accumulated large amounts of NR and NMN [60], which when present in culture media may act to protect these cells from mitochondrial damage and ROS accumulation [61]. These data suggest that in the experiments with primary cells the culture conditions already contained enough NAD+ precursors to maintain NAD+ levels, which remained at 80% [60] and 90% [59] of the level of growing cells in oral fibroblasts and IMR90, respectively. Therefore, NAD+ levels are maintained in fibroblast replicative senescence in the presence of adequate levels of NR. The IMR90 screen was conducted in physiological levels of glucose without pyruvate [59], whereas the oral fibroblast screen was conducted with high levels of glucose and pyruvate [60] and so none of the above changes is dependent on these variables. If the mouse diet is inadequate in NAD+ precursors as recent papers seem to indicate [61,63] and the normal healthy human diet is not, even in the elderly, this may explain why clinical NAD+ precursor trials have so far yielded only mild clinical benefit ([65]; reviewed in [66]. Consistently, increased PPP [59,60] would supply more NADPH [60] and so account for the decreased ROS levels observed in oral fibroblasts. However, the increase in NADPH is not specific to senescence as it occurs in quiescent and confluent cells as well [60].

### 2.5. Do Senescent Cells Exhibit a Reverse TCA Cycle?

Recently, we introduced the concept of the reverse TCA cycle with respect to cancer cells [69] and here we discuss the possibility that such a pathway might be operational in senescent cells. Senescent cells have been shown to have increased glutaminase 1 [74] conducting glutaminolysis (glutamine to glutamate and ammonia). The resulting ammonia antagonizes the acidic pH of senescent cells promoting their survival and eliminating glutaminase 1-dependent glutaminolysis ameliorated age-related pathologies in vivo [74]. In addition, glutaminolysis would increase the levels of glutamate and potentially activate the reverse cytoplasmic aconitase pathway. The activity of this enzyme would be sustained by high levels of intracellular iron associated with senescent cells [75] (see also below). Glutamate dehydrogenase would carry out the conversion of glutamate to alpha ketoglutarate with the concomitant reduction of NAD to NADH. Further, alpha ketoglutarate to isocitrate would result in an NADP+ increase. The result would be the overproduction of citrate in the cytoplasm through the reverse action of cytoplasmic aconitase. Glutamate and alpha-ketoglutarate could be also partially transported to the mitochondria (through aspartate glutamate carrier and 2-oxoglutarate carboxylase, respectively) changing the ratios between the intermediates and thus activating the reverse TCA cycle.

Although no NADP+ data were available in primary oral fibroblasts or IMR90, all non-proliferating oral fibroblasts have low NADH and a high NAD+/NADH ratio and increased NADPH [60] arguing against the reverse cytoplasmic aconitase pathway hypothesis in these cells. However, the hypothesis could be valid for senescent IMR90 cells as they do have a low NAD+/NADH ratio, high levels of intracellular citrate, very low levels of alpha ketoglutarate together with lower levels of fumarate, malate, and glutamate. The depletion of glutamate and alpha-ketoglutarate and the accumulation of citrate in IMR90 cells could be indicative of a reverse TCA cycle, which would produce citrate in the cytoplasm via cytoplasmic isocitrate dehydrogenase 1 (IDH1). IDH1 is an iron- and ROS-regulated enzyme [76] but when active can act as an antioxidant by the production of NADPH and alpha-ketoglutarate [77].

However, as emphasized above, dietary factors in both mice and humans together with cell culture conditions may influence phenotypes greatly and more work is required to test the above hypotheses.

Whilst it is recognized that the above arguments are based on the only two unbiased metabolomic screens of senescent cells published so far, several changes are common to both. These include an increase in metabolites involved in glycolysis, the PPP and redox homeostasis, and the differences between the screens may stimulate the investigation into such variables and the ability of cells to deal with oxidative damage and the availability of dietary factors. Furthermore, many studies on senescent cells employ at the most two cell lines [51,78] and frequently only one [79,80].

### 2.6. The Relationship of the Extracellular Senescence Metabolome to the Internal Senescence Metabolome

In the oral fibroblast screen above we reported that several metabolites specifically accumulated or became depleted in the conditioned medium of senescent fibroblasts on a per cell basis [70]. Whilst some of these changes could be attributed to increased senescent cell biomass two metabolites, citrate and C-mannosyl tryptophan remained significantly elevated in senescent fibroblast medium even when normalized to protein content. Extracellular levels of citrate (EC) relative to internal levels increased by over 20-fold in senescent fibroblasts when compared to growing, confluent or quiescent cells. Preliminary analysis of the effect of glucose and pyruvate suggests that lowering glucose to physiological levels has no effect on the levels of EC but eliminating pyruvate reduced the levels by about 50% and so pyruvate levels may have a considerable influence on senescent cell metabolism as pointed out previously [51].

## 3. The Role of p53 in Fibroblast Senescence and Metabolism

In classical senescence induced by genotoxic stress the levels of total p53, p53 phosphorylation, and transcription factor activity increase [81] and this restrains the SASP [26]. However, if DNA damage is not repaired within four or five days [26], cells become irreversibly cell cycle arrested and establish the senescent phenotype and the SASP. Eventually, p53 activity declines and levels of p16^INK4A^ rise [81]. This type of senescence is characterized by low chronic p53 and AMP activity, high NF kappa B activity and secreted HMGB1 (High mobility group box protein 1), and a high NAD+/NADH ratio [51]. Classical genotoxic SIPS results in a secretome involving IL-1 alpha and NF kappa B [51] and a shift in energy metabolism towards glycolysis [59,60,70].

In the absence of pyruvate, another form of senescence involving mitochondrial dysfunction and the depletion of sirtuins (SIRTs) 3 and 5 can be detected termed mitochondrial dysfunction-induced senescence (MiDAS). However, SIRTs 3 and 5 transcripts do not decline in all fibroblast lines following replicative senescence [82]. MiDAS is diametrically opposite to that induced by genotoxic stress and involves the secretion of a unique SASP involving AREG, IL-10, and CCL-27 [51], whilst suppressing the classical genotoxic stress-induced SASP via the upregulation of p53 and AMP [51].

MiDAS is dependent on mitochondrial DNA and antagonized by NAD+ and its precursors such as NMN [51], which dramatically accumulates in replicatively senescent cells in the presence of pyruvate [60]. NAD+ does not decline in senescent cells in the presence of adequate levels of its precursors [59,60] and pyruvate increases in the plasma and serum of humans with chronological age [83,84]. Therefore, it is unclear whether MiDAS has a dramatic effect during human aging although it may affect tissues that are unable to access or utilize pyruvate or be operational in age-related diseases.

### P53 Decline in Classical Genotoxic SIPS May Be Associated with a Shift towards Glycolysis and the PPP

The role of p53 in metabolism [85] and fibroblast senescence [56] has been reviewed recently and so will only briefly be covered here. Cells undergoing replicative senescence show increased levels of glycolytic and PPP enzyme transcripts and metabolites [70]. p53 antagonizes the uptake of glucose by lowering expression of the glucose transporters GLUT1 and GLUT4, transcriptionally activates the expression of TP53-inducible glycolysis and apoptosis regulator (TIGAR), which in turn dephosphorylates fructose-2, 6-bisphosphate to antagonize the early stages of glycolysis. p53 also binds glucose-6-phosphate dehydrogenase (G6PDH), inhibiting its activity and, consequently, the PPP and can also promote mitochondrial oxidative phosphorylation by inducing expression of synthesis of cytochrome c oxidase 2 (SCO2) and inhibiting pyruvate dehydrogenase kinase 2 (PDK2) through Parkin (PARK2) activation and so these last activities of p53 antagonize glycolysis and the PPP to promote mitochondrial respiration. Thus, the fall in p53 during the establishment of the senescence is likely to aid the shift towards glycolysis and away from oxidative phosphorylation in fully senescent primary oral fibroblasts where PDK2 transcript levels are slightly elevated [70]. In fact, most glycolytic and PPP metabolites and enzyme transcripts in such cells are elevated whilst most TCA metabolites are reduced [70] and similar observations have been made with embryonic lung fibroblasts undergoing proliferative exhaustion or following oncogene-induced senescence [59].

## 4. The Extracellular Senescence Metabolome and the Identification of Extracellular Citrate Accumulation as a Common Feature of Replicative Senescence and Senescence-Induced by Irreparable DNA Damage: Evidence for Cell Type Specificity

In addition to the unbiased analysis of the internal senescent fibroblast metabolome [60,70], we also analyzed the changes in metabolites in the conditioned medium of the same cells. As the initial objective was to identify metabolites that could potentially identify senescent cells in non-invasive tests, we normalized the data to cell number. We identified a total of 30 metabolites out of 261 that specifically accumulated in senescent fibroblast-conditioned medium relative to growing and cell cycle-arrested controls and 13 became specifically depleted [70]. Citrate and C-mannosyl tryptophan specifically accumulated relative to the controls independently of biomass and both have been implicated as markers of chronological age independent of other factors such as telomere length, sex, BMI, batch effect, and family relatedness [84].

C-mannosyl tryptophan (CMT) is thought to be epigenetically regulated and is associated with lung function, bone mineral density, and low birth weight in twins. Low birth weight in twins is linked to poorer health in later life and CMT may be an important indicator of human health span as well as chronological age [84]. It has been speculated that CMT is linked to apoptotic pathways, but this has not been confirmed experimentally and recent reports link it to a degradation product of autophagy [86]. CMT is increased in the urine in mouse models of diabetes [87] and in human kidney disease [88].

Citrate accumulated over 20-fold outside the senescent cells relative to the interior and relative to the controls and this is consistent with citrate being incompatible with glycolysis (see above) and requiring increased export in senescent cells. Citrate accumulates in IMR90, indicative of reduced activity of the ROS-sensitive mitochondrial aconitase ACO-2 [59] but senescent NHOF-1 show only a slightly reduced aconitate to citrate ratio indicative of low ROS. When five lines of fibroblasts from four tissue sites undergo either replicative senescence or SIPS due to irreparable DNA double-strand breaks, extracellular citrate (EC) accumulates up to 11-fold in the conditioned medium. EC accumulation is independent of biomass, repairable DNA damage, and cell cycle arrest and is associated with the establishment of senescence and the SASP rather than senescence initiation [89]. However, it is not yet clear whether citrate export increases or import declines.

Citrate is an important metabolite as it is not only required for the TCA cycle. After export from the mitochondria or import from the extracellular environment, citrate can serve as a substrate for ATP citrate lyase to produce acetyl CoA (reviewed in [90]). In turn, acetyl CoA serves as a substrate for lipid anabolism and cell growth [90] but also can participate in histone acetylation and epigenetic regulation [90].

The specific accumulation of EC and a shift to glycolysis following senescence is not observed in all cell types such as mammary epithelial cells undergoing proliferative exhaustion in the absence of irreparable DNA damage [91]. Therefore, epithelial cells may differ from fibroblasts in how they alter metabolism following the induction of senescence.

## 5. Citrate Transport and Its Role in Ageing and Disease

### 5.1. Citrate in the Diet and Its Potential Effect on Senescence and the SASP

The potential role of citrate in age-related pathologies is summarized in Figure 1. Citrate is also part of the diet and an important food preservative, which in conjunction with glucose induces the SASP in adipose tissue [92]. As part of an energy source, citrate is implicated in caloric restriction which can influence both senescent cell induction [93] and hence the SASP. Citrate is also present in very high concentrations as a blood transfusion product preservative and can induce inflammatory cytokines in macrophages [94]. Therefore, there is evidence that citrate can induce inflammatory cytokines prior to detectable age-related disease.

### 5.2. Citrate Import by SLC13A5/I’m Not Dead Yet (INDY)

*SLC13A5/INDY* was originally identified as a regulator of lifespan in *Caenorhabditis elegans* lifespan [95] and later in *drosophila melanogaster* [96] by mechanisms mimicking and interacting with caloric restriction. Mammalian SLC13A5/INDY protein is a sodium-dependent transporter of citrate and to a lesser extent succinate, malate, or fumarate [97,98,99]. Systemic and targeted knockout or knockdown of the mouse homolog of *SLC13A5/INDY* (*mindy*) in the liver reveals that *mindy* mediates type 2 diabetes and several characteristics of the age-related cardiometabolic syndrome including adiposity, non-alcoholic fatty liver, insulin resistance, and mitochondrial dysfunction [100]. In addition, *mindy* is expressed in the adrenal medulla and *mindy* knockout mice show lowered levels of citrate, catecholamines, and significantly, heart rate and blood pressure [101]. These studies underline the link between citrate uptake, obesity, type 2 diabetes, and blood pressure. In addition, mouse *mindy* is expressed in neurons and systemic deletion of the *mindy* gene significantly improves memory performance and motor coordination of mice. Tissue-specific knockout of *mindy* in the neurons resulted in improved memory whereas liver-specific deletion did not [102] and neither targeted deletion affected motor-co-ordination, suggesting that *mindy* may have a role in other tissues affecting the last phenotype. The improved memory of *mindy*-KO mice was associated with increased hippocampal neurogenesis and dendritic spine formation in dentate granule cells, which are well-documented contributors to enhanced memory performance [102]. This is particularly interesting given that changes in the hippocampus have been linked with early signs of Alzheimer’s disease [103]. Drugs targeting SLC13A5/INDY are now available and are reported to work in vitro [101] and in vivo [104]. Furthermore, bone is the main site of citrate storage in mammals, and bone demineralization is likely to release citrate in individuals with osteoporosis, which has recently been linked with dementia in humans [105,106].

However, these exciting findings should be tempered with caution. Firstly, it should also be noted that human SLC13A5/INDY has a much lower affinity for citrate than its mouse counterpart [107,108] and loss of function mutations of *SLC13A5/INDY* in humans leads to severe disease such as early infantile epileptic encephalopathy-25/developmental epileptic encephalopathy-25 [109,110] reviewed in [107,108]. Brain citrate levels are high compared to blood and other soft tissues and can reach up to 400 μM in cerebrospinal fluid [111]. This increased level of citrate in the brain is determined to be sustained by astrocytes. The exact mechanism of citrate release from astrocytes remains to be determined. Citrate in the brain is believed to control neuronal excitability through divalent cations chelation [111]. Interestingly, Li^+^, known to enhance citrate uptake through SLC13A5/INDY expressed in neurons [112], is used to treat bipolar disease and has been shown to have a neuroprotective role [113]. This is consistent with the importance of citrate uptake by neurons and changes in citrate supply from astrocytes and/or citrate uptake by neurons could potentially contribute to neurodegenerative diseases. It has been suggested that the differences in the mouse (*mindy*) and human (*SLC13A5/INDY*) phenotypes could be due to the blood–brain barrier [108], but recent experiments with targeted *mindy* knockout in the nervous system argue against this [102]. Moreover, the injection of citrate into the brain also causes epilepsy [107] and so too much citrate can be just as detrimental as too little. In addition, *SLC13A5/INDY* expression is strongly associated with obesity, insulin resistance, and fatty liver in humans [114]. These issues will only be resolved when SLC13A5/INDY inhibitors are approved for testing in humans in clinical trials. In humans, SLC13A5/INDY has also been found to be highly expressed in the testes [107] where citrate could be used to support sperm. Citrate is synthesized and released by prostate epithelial cells and released into prostatic fluid where its level can be as high as 180 mM. In seminal fluid, the level of citrate was found to vary between 13–50 mM [115]. Prostate secretory epithelial cells release citrate through pmCiC [116] and changes in citrate content in the prostatic fluid could contribute to decreased fertility.

Citrate is also imported by the bidirectional citrate transporter pmCiC, which is a variant of SLC25A1 directed to the plasma membrane instead of the mitochondrial membrane. This is due to *SLC25A1* and pm*CiC* having alternative first exons [116]. pmCiC is interesting because it can act as a citrate exporter in normal cells but appears to revert to importing citrate in cancer cells and is upregulated in a wide variety of cancer tissues [117]. The addition of a pmCiC inhibitor can reduce citrate import in cancer cells, alter metabolism, and inhibit pancreatic carcinoma xenograft growth in vivo [117]. pmCiC is expressed in both the cancer cells and the surrounding mesenchyme [118] and citrate derived from cancer-associated fibroblasts has recently been shown to modulate the cytokine profile, epithelial–mesenchymal transition, mesenchymal–epithelial transition, and the metastatic phenotype [118]. Therefore, citrate can act in a paracrine manner to stimulate tumor growth and progression. Cancer cells can also regulate the production of extracellular citrate in cancer-associated fibroblasts and change their cytokine profile to aid tumor progression [118].

### 5.3. Citrate Export and Its Upregulation in Senescence and Fibroblast Activation

We have recently reviewed the role of citrate in the cancer environment [67,69]. In addition, senescent cells have been implicated in a wide range of age-related pathologies including several where the import of citrate has been implicated such as type 2 diabetes [100,119], cardiovascular disease [101,120], memory [102,121], and cancer [67,69,117,118,122]. Furthermore, the injection of even a small number of senescent pre-adipocytes [123] or adipose-derived mesenchymal stem cells from old donors [124] causes numerous age-associated deleterious phenotypes and reduces the lifespan in mice.

Prostate epithelial cells secrete high levels of citrate which is thought to aid the energy requirements of sperm [125]. The plasma membrane citrate transporter pmCiC is expressed at very high levels in human benign prostatic hyperplasia [125]. Interestingly benign prostatic epithelia share many characteristics with those of senescent and activated fibroblasts such as increased levels of senescence-associated beta-galactosidase and transforming growth factor-beta (TGF-β) [126,127,128]. Another plasma membrane transporter encoded by the *ANKH* gene is also expressed in the prostate and has recently been shown to export citrate in addition to malate, succinate, and phosphate [129]. Our preliminary analysis of pmCiC is that it is not consistently upregulated during replicative senescence or SIPS but may be upregulated in some lines by TGF-β, which is a major inducer of fibroblast activation. However, our knowledge of plasma membrane transporters and their role in the transport of TCA metabolites is far from complete and there are considerable gaps in our knowledge of the mechanism of EC accumulation following senescence.

## 6. Conclusions

There is now compelling evidence that senescent cells are instrumental in a wide range of age-related human pathologies and there is evidence that senescent cells can cause these pathologies in mice. One of the proposed mechanisms is the induction of an array of proteins collectively referred to as the SASP. One of the most popular hypotheses supported by a good body of evidence is that the SASP is induced by inflammatory signals mediated by extranuclear chromatin, irreparable DNA double-strand breaks, and dysfunctional mitochondria. However, there is insufficient evidence to conclude that senescent cells or the SASP cause aging as opposed to age-related disease. Furthermore, there is evidence to suggest that caloric intake can determine the induction of senescence and the SASP. Citrate is such a source of calories, increases in the plasma of humans with chronological age [83,84] and is an external energy source for many cell types. The potential role of citrate in age-related pathologies is summarized in Figure 1. In addition, EC accumulates in the extracellular environment of at least some senescent cell types and citrate import parallels that of senescent cells in many age-related pathologies. Moreover, citrate in concert with glucose can induce early indicators of both type 2 diabetes and Alzheimer’s disease in mice, supporting a potential causal role for citrate and that of energy intake in aging. Whilst much further investigation is required, it seems plausible that EC is an important component of the SASP of several cell types and highly relevant to aging and age-related disease.

## Figures and Tables

**Figure 1 ijms-23-03652-f001:**
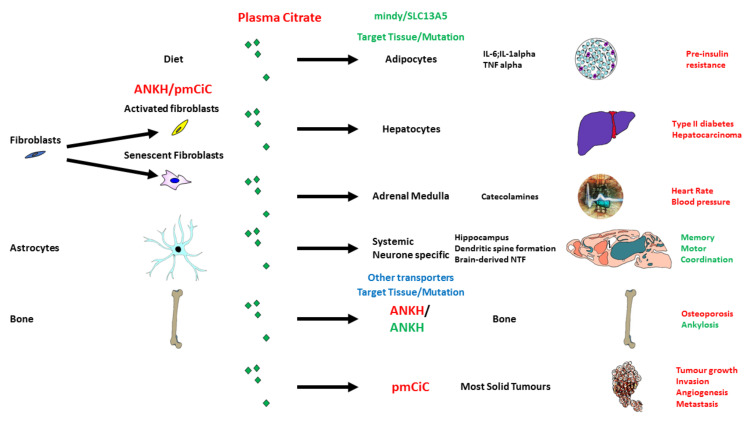
Cartoon summarizing the potential role of citrate in aging and age-related diseases. The cartoon summarizes evidence that citrate either supplied via the diet or derived from certain types of senescent cells can potentially be instrumental in age-related diseases such as type 2 diabetes, increased blood pressure, memory loss, and cancer. Plasma citrate is indicated by the green diamonds, gained functions are in red and reduced functions are in green. In the case of *ANKH* red is gain of function mutations in humans and green loss of function mutations as well as mouse knockout. Green in *SCL13A5*/*mindy* indicates results from the mouse knock out.

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
