# Peer review of "Metabolic Alterations in Cellular Senescence: The Role of Citrate in Ageing and Age-Related Disease"

_ijms, 2022, doi:10.3390/ijms23073652_

Round 1

Reviewer 1 Report

In this manuscript, the authors performed meta-analysis using the published data. They re-analyzed the metabolite analysis data in the growing cell and senescent cells. They intended to draw the significant data, however the datasets, which were used in this study were very limited. Therefore the analysis is very preliminary, and also not significant. For this reason, this manuscript is not suitable for the publication of IJMS.

The title was (the role of citrate…), however the description for this topic is less than 10 lines (421-429).

Figure 1~4. The analysis used only 2 cell line data, and they should collect more cell line data to support the authors’ claim.

The references should be updated, and most references were published before 2015. Thus the review contain outdated information.

The introduction contain very old information, and the authors should update the information.

Author Response

In this manuscript, the authors performed meta-analysis using the published data. They re-analyzed the metabolite analysis data in the growing cell and senescent cells. They intended to draw the significant data, however the datasets, which were used in this study were very limited. Therefore the analysis is very preliminary, and also not significant. For this reason, this manuscript is not suitable for the publication of IJMS.

Answer: The article was intended to be a review article supplemented with data from the only two metabolomic screens of human fibroblast senescence that have been published.

Firstly, the intention was not to necessarily demonstrate commonality but the point out key differences between the two studies and highlight the potential variables that contribute to differences which we believe will be very important when attempting to translate work to the clinic. Recent data by Sun et al EMBO J. 2020 in particular has emphasised the importance of nutritional variables in the regulation of the senescent phenotype.  In the absence of an appreciation of these types of variables bioinformatics analyses on existing data sets may turn out to be flawed.

Secondly, most cell and molecular biology studies in the senescence field conduct work on at most two fibroblast lines and so whilst we agree in principle with the referee unless more groups perform unbiased metabolomic screens of senescent cells this is all the data that is available.

Nevertheless, we have removed Figures 1-6 and merely cited and discussed them. See our response to Referee#2.

The title was (the role of citrate…), however the description for this topic is less than 10 lines (421-429).

Answer: The title was ‘Metabolic alterations in cellular senescence: the role of citrate in ageing and age-related disease’ We needed to set the scene for the age-related disease part as most readers still do not appreciate that citrate is an important energy source which in turn is an established driver of both cellular senescence (Fontana et al Ageing Cell 2018). Citrate was referred to exclusively in two sections  out of five (4 and 5) of the original article and now represents lines 274 to 286 and 331 to 496 of the revision, which is 34% of the article plus Figure 1. It is also extensively mentioned in the conclusions.

Figure 1~4. The analysis used only 2 cell line data, and they should collect more cell line data to support the authors’ claim. Answer: See above.

The references should be updated, and most references were published before 2015. Thus the review contain outdated information.

Answer: The metabolism field is very old and still not widely investigated in cellular senescence (see Wiley et al Cell Metabolism 2016). Similarly, the biology of the senescent phenotype was largely investigated in the 1990s. Where the field has moved on via failed telomere repair and the characterisation of the SASP we have cited up to date references.

 If the referee has specific examples, we would be only too willing to include them.  There is no point in quoting recent references when they add nothing to the original paper’s conclusions. However, we have cited a new reference 10 to support the 2001 reference of Munro et al Oncogene 2001.

Furthermore, 70% of the cited references were published after 2010 and so the reference list is hardly outdated.

The introduction contain very old information, and the authors should update the information.

Answer: See above

Reviewer 2 Report

I must admit, that this review looks quite strange, as if it was a mix of a research article and a “normal” review. Indeed, 6 out of 7 figures are Excel charts, comparing different metabolites in 2 cells lines. It looks as if the authors did not know what to do with a chunk of experimental data and decided to introduce them in the middle of a small review to extend the volume of otherwise quite short manuscript. Indeed, the same group of authors have just published quite big review (not cited in the current manuscript) about the role of the extracellular citrate in cancer cells ( Parkinson et al., Cancer and Metastasis Reviews, published online 21 December 2021). The chapter 8 of this review called “Citrate Export and its upregulation in Senescent fibroblast” – compare to chapter 5.3 “Citrate Export and its Upregulation in Senescence and fibroblast activation”. I strongly suggest to re-consider the Figures 1-6 or better delete them at all or transform to some kind of a Table. The same is for the chapter 2.5, which looks like a Discussion of experimental paper.

There is nothing wrong with publishing small reviews. However, it is not quite appropriate to introduce the experimental data (6 Figures!) inside of a review. So, it must be re-written.

Minor remarks:

  • I did not understand why in the Figure 7 “ANKH/pmCiC, plasma citrate” are written in red and “mindy/SLC135, Target/Tissue/mutation” is in green. In the Figure legend it is mentioned that gained functions are in red and reduced functions are in red”, but it has nothing to do with the color-coding of titled of columns in the Table..

  • It would be better if the Chapter 1.3 would move before the actual 1.2 (Telomeres and Senescence” or at the end of the Introduction. Currently it this chapter is inserted in the block about telomeres (chapters 1.2, 1.4, 1.5) and this position cuts the flowchart.

Author Response

I must admit, that this review looks quite strange, as if it was a mix of a research article and a “normal” review. Indeed, 6 out of 7 figures are Excel charts, comparing different metabolites in 2 cells lines. It looks as if the authors did not know what to do with a chunk of experimental data and decided to introduce them in the middle of a small review to extend the volume of otherwise quite short manuscript.

Answer: Indeed, the same group of authors have just published quite big review (not cited in the current manuscript) about the role of the extracellular citrate in cancer cells ( Parkinson et al., Cancer and Metastasis Reviews, published online 21 December 2021). The chapter 8 of this review called “Citrate Export and its upregulation in Senescent fibroblast” – compare to chapter 5.3 “Citrate Export and its Upregulation in Senescence and fibroblast activation”. I strongly suggest to re-consider the Figures 1-6 or better delete them at all or transform to some kind of a Table. The same is for the chapter 2.5, which looks like a Discussion of experimental paper.

Answer: We agree with the referee.  We have removed the experimental data completely. We could not cite the Cancer Metastasis Reviews article because at the time we do not think it had even been reviewed when we submitted the current manuscript and was certainly not in press.

We have now cited this review where appropriate (reference 72; lines 243 and 450) as it introduces the concept of a reverse TCA cycle and this could be applied to senescent cells.

There is nothing wrong with publishing small reviews. However, it is not quite appropriate to introduce the experimental data (6 Figures!) inside of a review. So, it must be re-written.

We have removed Figures 1-6 and merely cited and discussed them as all the data has been published.

We have deleted the detailed discussion in section 2.5 and quoted references instead as in a conventional review as the work has been published and the data sets are available.

We have split section 2.5 into two and created a new section 2.6 (lines 241-272) which is devoted to introducing the concept of cited our recent review (reference 72) in the appropriate sections.

Minor remarks:

  • I did not understand why in the Figure 7 “ANKH/pmCiC, plasma citrate” are written in red and “mindy/SLC135, Target/Tissue/mutation” is in green. In the Figure legend it is mentioned that gained functions are in red and reduced functions are in red”, but it has nothing to do with the color-coding of titled of columns in the Table..
  • Answer: There are different phenotypes for ANKH gain of function and loss of function mutations but have expanded the Figure 1 legend to explain this better (Lines 377-379)

  • It would be better if the Chapter 1.3 would move before the actual 1.2 (Telomeres and Senescence” or at the end of the Introduction. Currently it this chapter is inserted in the block about telomeres (chapters 1.2, 1.4, 1.5) and this position cuts the flowchart.
  • Answer: We have moved the position of section 1.3. The new 1.2 is now lines 49-67 and 1.3 lines 68-103.

Round 2

Reviewer 1 Report

In this revised manuscript, the authors removed figure 1-6 based on other reviewer’s comment. When I reviewed this manuscript previously, due to the insufficient literature and data, I did not think this manuscript was suitable for the publication of IJMS. In this review, the authors mainly reivewed two published articles. Althouh the authors removed all the figures, their data were inconsistent. Thus, more data and publications will be required to publish the review paper. I think it is premature to publish the review based on the handful of data.

Author Response

Referee#1 complained about the two metabolic screens and comparing the two not about citrate specifically. We have now acknowledged this on lines 273-280 of the revised manuscript but also pointed out that most studies in the senescence field use at best two lines and some investigate only one (usually IMR90). I have sent Lucy Zhang pdfs of these papers in high impact journals. Therefore, it is grossly unfair to criticise our review on the grounds that we have discussed the similarities and differences between the only two unbiased metabolomic screens published. Furthermore, in effect we are highlighting the issue raised by referee#1.

If extracellular citrate in meant both the referee#1 and the academic editor should read the paper by James et al JPR 2015 including the supplementary information where we have shown extracellular citrate to be upregulated by replicative senescence in five fibroblast lines from four different sites (oral, skin, lung and colon) and seven lines following ionizing radiation induced senescence (a common protocol for inducing senescence). The accumulation of EC

in senescence is specific regarding inflammation, repairable DNA  damage and cell cycle arrest and independent of biomass. Unfortunately, we did not emphasize the reproducibility of this observation in the review because we would have expected an expert reviewer to be aware of it as it has been cited 115 times and this may have led to the original confusion.

We have now made this much clearer in lines 357-362 of the revised manuscript.

The elevation of citrate with chronological age (the second part of the review and in the title) has been shown in studies amounting to tens of thousands of people (Menni et al Int.J.Epidemiol. 2013;Auro et al Nature Communications 2014) and functional significance determined in SLC13A5/INDY knockout mice and human cancer xenografts.

It is very odd that IJMS are perfectly happy for me to review the data in Wiley and Campisi which is based on studies conducted on one or at best two cell lines but find our work on extracellular citrate insufficient; this does not make sense. I have further in vivo data in humans supporting our case but cannot divulge this until at least one of the papers is in press.

Reviewer 2 Report

The authors addressed all my comments. The review can be published in present form.

Author Response

No response required but see response to referee#1 and the academic editor.